

# Application of prohexadione-calcium priming affects *Brassica napus* L. seedlings by regulating morph-physiological characteristics under salt stress

Peng Deng[1], Aaqil Khan[1], Hang Zhou[1], Xutong Lu[1], Huiming Zhao[1], Youwei Du[1], Yaxin Wang[1], Naijie Feng[1,2,3] and Dianfeng Zheng[1,2,3]

[1] College of Coastal Agricultural Sciences, Guangdong Ocean University, ZhanJiang, GuangDong, China
[2] Shenzhen Institute, Guangdong Ocean University, Shenzhen, Guangdong, China
[3] South China Center of National Saline-tolerant Rice Technology Innovation Center, South China, Zhanjiang, Guangdong, China

Corresponding authors
Naijie Feng, fengnj@gdou.edu.cn
Dianfeng Zheng,
zhengdf@gdou.edu.cn

## ABSTRACT

Salinity stress imposes severe constraints on plant growth and development. Here, we explored the impacts of prohexadione-calcium (Pro-Ca) on rapeseed growth under salt stress. We designed a randomized block design pot experiment using two rapeseed varieties, 'Huayouza 158R' and 'Huayouza 62'. We conducted six treatments, S0: non-primed + 0 mM NaCl, Pro-Ca+S0: Pro-Ca primed + 0 mM NaCl, S100: non-primed + 100 mM NaCl, Pro-Ca+S100: Pro-Ca primed + 100 mM NaCl, S150: non-primed + 150 mM NaCl, Pro-Ca+S150: Pro-Ca primed + 150 mM NaCl. The morphophysiological characteristics, and osmoregulatory and antioxidant activities were compared for primed and non-primed varieties. Our data analysis showed that salt stress induced morph-physiological traits and significantly reduced the antioxidant enzyme activities in both rapeseed varieties. The Pro-Ca primed treatment significantly improved seedlings, root, and shoot morphological traits and accumulated more dry matter biomass under salt stress. Compared to Huayouza 158R, Huayouza 62 performed better with the Pro-Ca primed treatment. The Pro-Ca primed treatment significantly enhanced chlorophyll content, net photosynthetic rate ($Pn$), stomatal conductance ($Gs$), transpiration rate ($Tr$), and actual photochemical quantum efficiency ($\Phi PSII$). Furthermore, the Pro-Ca primed treatment also improved ascorbic acid (ASA) content, superoxide dismutase (SOD), peroxidase (POD), catalase (CAT), and ascorbate peroxidase (APX) activity, and stimulated the accumulation of soluble proteins. These findings strongly suggested that the Pro-Ca primed treatment may effectively counteract the negative impacts of salinity stress by regulating the morph-physiological and antioxidant traits.

## INTRODUCTION

Soil salinity is a serious environmental problem that has detrimental impacts on agricultural production across the world. It results in low crop yield and poor grain quality.

Approximately $3.4 \times 10^8$ hm$^2$ of land is adversely affected by salt, which accounts for about 23% of total arable land globally (*Wang et al., 2019*). The salt-affected area in China is around $9.2 \times 10^7$ hm$^2$, equivalent to 5% of the total land area (*Zhang & Wang, 2021*).

Rapeseed is one of the most vital oil-seed crops that supply consumption products such as meal protein for humans and animal feed (*Sabbahi et al., 2023*). Salinity stress impacts rapeseed yield by reducing seed germination, seedling growth and development, plant vigor, grain yield, and quality of the rapeseed (*Borchers & Pieler, 2010*). Excessive Na$^+$ and Cl$^-$ ions produce toxic components that can impair plant growth and development. These ions generate toxicity, oxidative stress, and induce osmotic potential in the cells (*Tian et al., 2022*). Salt stress also causes oxidative stress in rapeseed leading to increased levels of reactive oxygen species (ROS), which affect cellular metabolism and cause physiological disorders (*El-Badri et al., 2022*). Plant cells regulate ROS levels through an antioxidant metabolic homeostatic defense system containing enzymatic and non-enzymatic antioxidants (*Chiappero et al., 2021*). The main antioxidant enzymes that stimulate cellular oxidative homeostasis are superoxide dismutase (SOD), peroxidase (POD), and catalase (CAT) (*Dong et al., 2023*). Non-enzymatic antioxidants such as ascorbic acid (ASA) and glutathione (GSH) are responsible for eliminating ROS and alleviating the oxidative damage caused by salt stress (*Kaya et al., 2020*).

Seed priming is a low-cost, simple, and effective strategy that promotes seed germination, the neatness of seedling emergence, lower external water absorption, and the development of solid seedlings (*Silva et al., 2023*). Seed priming techniques may enhance salt tolerance in plants and are also cost-effective and environmentally-friendly (*Johnson & Puthur, 2021*). Many crops have been shown to profit from seed priming in terms of seed germination, seedling establishment, and eventually, crop productivity under adverse environmental situations (*Juanna et al., 2016*). Seed initiation has been applied to rapeseed to alleviate the damage suffered by rapeseed under salt stress, improve the seed germination and photosynthetic capacity of seedlings, and maintain higher viability of rapeseed under salt stress (*El-Badri et al., 2022*).

Plant growth regulators are largely used in controlling plant growth and development (*Desta & Amare, 2021*). Prohexadione-calcium (Pro-Ca) is a class of environmentally-friendly plant growth regulators that increase crop yield quality (*Chang, 2016*) and improve plant tolerance to salt stress (*Huang et al., 2023*). Pro-Ca may control wilt by thickening the cell walls of the thin-walled tissues of the cortex (*Wallis & Cox, 2020*). Some scholars have shown that spraying Pro-Ca on the leaves of rice under salt stress enhances the resistance of rice seedlings to collapse and improves the salt tolerance of rice seedlings by increasing their photosynthetic and antioxidant capacity (*Zhang et al., 2023*). Other studies have shown that Pro-Ca can effectively reduce the damage caused by salt stress and improve the salt tolerance of soybean seedlings by regulating the photosynthetic capacity, antioxidant defense capacity, and osmoregulation capacity of soybean seedlings (*Feng et al., 2021*). Therefore, Pro-Ca has the potential to improve the salt tolerance of crops and may greatly benefit the cultivation of saline-alkaline crops.

The present study was intended to determine the mechanisms of Pro-Ca seed priming on rapeseed germination, seedling growth and development, and morph-physiological

traits under salt stress. The Pro-Ca primer may reduce the injury of rapeseed under salt stress and promote its growth and development. This study also explored the regulatory mechanisms of Pro-Ca primers on physiological traits such as photosynthesis and antioxidant defense system in rapeseed under salt stress. The purpose was to study the effect of Pro-Ca elicitation on the salt tolerance of rapeseed, to provide new ideas for improving the salt tolerance of rapeseed, and to provide a basis for cultivating rapeseed in saline and alkaline soils.

# MATERIALS AND METHODS

## Plant materials and pharmaceuticals

Rapeseed varieties Huayouza 158R and Huayouza 62, were used as plant material in this study. The plant growth regulator Pro-Ca (8 mg $L^{-1}$) was used for seed induction. Both materials were supplied by College of Coastal Agricultural Sciences, Guangdong Ocean University.

## Experimental design

The experiment was conducted in a completely randomized block design. The study was conducted from 2022–2023 in a daylight-linked greenhouse (natural light conditions, day/night temperature difference of 25/20 ± 2 °C, relative humidity of 60%) at the Binhai College of Agriculture, Guangdong Ocean University (latitude: 21°8′56″N, longitude: 110°17′58″E, altitude: 20 m). Uniform and mature seeds were manually selected and were sterilized with 3% hydrogen peroxide for 10 min, then thoroughly rinsed three times with distilled water. The dried seeds were subjected to the priming treatment, which consisted of 10 ml of Pro-Ca at a concentration of 8 mg $L^{-1}$. The solution was used to moisten petri dishes lined with two layers of filter paper, and the sterilized seeds were evenly placed into the petri dishes, sealed with an airtight film and primed for 8 h in an incubator under darkness at 20 °C. The size of the germination pot was 19 cm × 14 cm × 17 cm and the soil substrate was a 3:1 mix of nutrient soil and sand. Six holes were made in the soil of each pot and three seeds were sown in each hole, totaling 18 seeds. When the plant developed three true leaves, the plants were thinned leaving one plant in each hole for a total of six plants. The treatments of various concentrations of NaCl (0, 100, 150 mM) were set up in the experiment. A quantity of sodium chloride was dissolved into 1,000 ml of water per pot for the salt treatment and this solution was mixed well with the soil. Sampling was done 21 days after sowing. The various concentrations of seed priming included:

    (1) S0: non-primed + 0 mM NaCl.
    (2) Pro-Ca+S0: Pro-Ca primed + 0 mM NaCl.
    (3) S100: non-primed + 100 mM NaCl.
    (4) Pro-Ca+S100: Pro-Ca primed + 100 mM NaCl.
    (5) S150: non-primed + 150mM NaCl.
    (6) Pro-Ca+S150: Pro-Ca primed + 150 mM NaCl.

## Determination of above-ground morphological indicators and biomass

Representative rape seedlings of each treatment were selected and rinsed with distilled water. Plant height was measured with a ruler, stem thickness was measured with a vernier caliper, and leaf area was scanned with a Yaxm-1241 leaf area meter (Beijing Yaxin RIYI Technology Co., Ltd., Beijing, China). The rapeseed plants were baked in an oven at 105 °C for 30 min, after which the temperature was adjusted to 80 °C and the leaves were dried to a constant weight. An electronic balance was used to weigh the dry weight of the above and below ground parts of the plant.

The seedling strength index was calculated by referring to the method of *Liu et al. (2015)*.

Seedling strength index = (Sub-ground dry weight/dry weight above ground + stem thickness/plant height) (Sub-ground dry weight + dry weight above ground).

## Measurement of root morphological indexes

Twelve representative seedlings of each treatment were selected and washed with distilled water, and the root system of each seedling was cut and scanned by a desktop scanner (Epson CORP, Suwa, Nagano, Japan). The images were analyzed with WinRHIZO root analysis software (Regent Instruments, Quebec, Canada) to obtain relevant indexes such as total root length, total root surface area, total root volume, and total root tip number.

## Measurement of leaf photosynthetic indexes

A SPAD-502 chlorophyll meter was used to determine the relative chlorophyll content (SPAD) of functional leaves of rapeseed. Gas exchange parameters, including net photosynthetic rate ($Pn$), intercellular $CO_2$ concentration ($Ci$), transpiration rate ($Tr$), actual photochemical quantum efficiency ($\Phi PSII$), and stomatal conductance ($Gs$) were measured from 9:00 to 11:30 a.m. using a portable photosynthesizer LI-6800 (LI-COR, Inc., Lincoln, NE, USA). The measurements took place under the following conditions: light intensity of 1,000 µmol m$^{-2}$ s$^{-1}$, $CO_2$ concentration of 400 µmol mol$^{-1}$, leaf temperature of 25 °C, relative humidity at 60–70%, and airflow rate of 500 µmol s$^{-1}$.

## Cell damage and membrane stability

The thiobarbituric acid method was used to determine the malondialdehyde (MDA) content by homogenizing frozen leaf samples (500 mg) in 10 mL of 10% trichloroacetic acid (TCA), centrifuging at 6,000 g for 20 min, and aspirating the supernatant for its determination. The absorbance of the reaction solution was measured at 450, 532, and 600 nm, respectively. The MDA content was calculated according to Anastasiou's formula (*Anastasiou et al., 2014*). Membrane stability was assessed by absolute conductivity, which was measured by weighing 0.1 g of fresh leaves soaked in 10 mL of deionized water at room temperature for 24 h, measuring conductivity (R1) with a conductivity meter, water bathing in boiling water for 30 min, and measuring conductivity (R2) after cooling. The absolute electrical conductivity (EL) of the leaves was calculated according to: EL = R1/R2 × 100% (*Li et al., 2014*).
## Determination of antioxidant enzyme activity and non-enzymatic anti-oxidant content

The antioxidant enzyme activity of the leaves was determined by weighing 0.5 g of frozen leaf samples and grinding them into a homogenate in 10 mL of 0.05 mol L$^{-1}$ pre-chilled phosphate buffer (pH 7.8) then centrifuging at 12,000 g for 20 min at 4 °C. The supernatant was used to determine superoxide dismutase (SOD), peroxidase (POD), catalase (CAT), and ascorbate peroxidase (APX) activities. SOD activity was determined by the method of Giannopolitis, where the change in absorbance at 560 nm was measured. One unit of SOD activity was defined as the amount of enzyme that inhibits 50% of NBT photoreduction (*Giannopolitis & Ries, 1977*). POD activity was determined by the oxidation rate of guaiacol at 470 nm (*Zhang et al., 2015*); CAT activity was determined by the decrease in absorbance per minute at 240 nm due to H$_2$O$_2$ consumption by Maehly's method (*Chance & Maehly, 1954*); APX activity was determined by the change in absorbance per minute at 290 nm by *Ahanger et al. (2018)* method for ascorbate. Phenylalanine ammonia-lyase (PAL) activity was determined by *Gholizadeh & Kohnehrouz (2010)* method (*Ahanger et al., 2018*), by taking 0.5 g of frozen sample, placing it in a pre-chilled mortar, adding 10 mL of 0.1 mol L$^{-1}$ borate buffer (pH 8.8, containing 0.1% mercaptoethanol), then adding 0.5 g of PVP and a little quartz sand, and centrifuging the supernatant at 4 °C and 15,000 g for 20 min. Min and the absorbance value of the supernatant were measured at 290 nm (*Gholizadeh & Kohnehrouz, 2010*).

The contents of ascorbic acid (ASA) and glutathione (GSH) were determined concerning the method of *Huang et al. (2005)* and *Ahmad et al. (2020)*. A total of 0.5 g of rapeseed leaves were taken, 10 ml of 5% TCA solution was added, ground at 4 °C, centrifuged at 20,000 g for 15 min, and the supernatant was used to determine the content of ASA (*Huang et al., 2005*). Plant leaves weighing 1.0 g were added to 5 ml of 5% metaphosphoric acid solution, ground at 4 °C, and centrifuged at 20,000 g for 15 min; the supernatant was used to determine the content of GSH (*Ahmad et al., 2020*).

## Measurement of osmoregulatory substances

The method described by *Kučerová et al. (2019)* was used to determine the soluble protein content of the samples. A total of 0.5 g of the fresh leaves was weighed and ground into a homogenate in 10 mL of 0.05 mol L$^{-1}$ pre-chilled phosphate buffer (pH 7.8), transferred to a centrifuge tube and centrifuged at 4 °C and 12,000 g for 20 min. The resulting supernatant was the protein crude extract which was used for the determination of soluble protein content. To determine the soluble sugar content, we weighed 0.1 g of frozen sample of leaves, then put them into a centrifuge tube containing 4 mL of 80% ethanol solution and ground it into a homogenate; this was extracted in a water bath at 80 °C for 20 min, removed and centrifuged at 4,000 g for 5 min. The supernatant was collected, and this process was repeated three times to fix the volume to 25 mL. The aspirated extracts were measured according to *Wang et al. (2022)*.

## Statistical analysis

All data were analyzed by one-way ANOVA using SPSS 25.0. Duncan's test was used to evaluate the difference among different treatments; $P < 0.05$ indicated that there was a statistical difference between treatments. Origin 2021 software was used for plotting.

## RESULTS

### Effect of Pro-Ca priming on the growth of rapeseed seedlings under salt stress

In this study, NaCl stress significantly affected all growth parameters of rapeseed seedlings. Plant height, stem thickness, leaf area, dry weight, and seedling strength index were significantly reduced with increasing salt stress (Figs. 1 and 2). The dry weight of Huayouza 158R decreased by 55.0% and 75.1% when salt stress was at S100 and S150, respectively. The dry weight of Huayouza 62 decreased by 57.1% and 61.0% when salt stress was at S100 and S150, respectively (Fig. 2D).

Pro-Ca treatment significantly reduced the inhibitory effect of NaCl stress on rapeseed seedlings. The addition of the Pro-Ca primed treatment (Pro-Ca+S0) under S0 conditions promoted seedling growth (Fig. 1). The addition of Pro-Ca primed treatment (Pro-Ca+S100) under S100 salt stress significantly increased plant height, stem thickness, leaf area, dry weight, and seedling strength index (Fig. 2). The addition of Pro-Ca primed treatment (Pro-Ca+S150) under S150 salt stress increased leaf area and seedling strength index by 30.59% and 12.26% in Huayouza 158R and 31.00% and 34.11% in Huayouza 62, respectively (Figs. 2C and 2D). The results showed that Pro-Ca treatment did not have the same promotion effect on the two varieties, and the promotion effect on Huayouza 62 was better, which is meaningful for the development and utilization of saline soils (Figs. 1 and 2).

### Effect of Pro-Ca priming on root growth of rapeseed under salt stress

The root development of rapeseed was inhibited with increasing levels of salt stress (Figs. 1 and 3). From S0 to S150, the total root surface area, total root mean diameter, total root length, total root volume and total root tip number of Huayouza 158R decreased by 62.80%, 14.70%, 57.03%, 67.09% and 51.57%, respectively, with the increase of salt stress level. Huayouza 62 decreased by 54.63%, 7.19%, 54.06%, 55.81%, and 46.51%, respectively, under the same conditions (Fig. 3).

Compared with salt stress, Pro-Ca induced root growth and significantly increased all indexes except the average total root diameter. With Pro-Ca+S100, the total root surface area, total root length, total root volume and total root tip number of Huayouza 158R increased by 175.32%, 235.08%, 134.04%, and 276.28%, respectively. Huayouza 62 increased by 235.89%, 243.44%, 268.42%, and 258.39%, respectively (Figs. 3A, 3C, 3D and 3F). With Pro-Ca+S150, total root surface area, total root length, total root volume, and total root tip number increased by 130.05%, 93.00%, 200.00%, and 102.39% in Huayouza 158R, respectively. Total root surface area, total root length, total root volume, and total

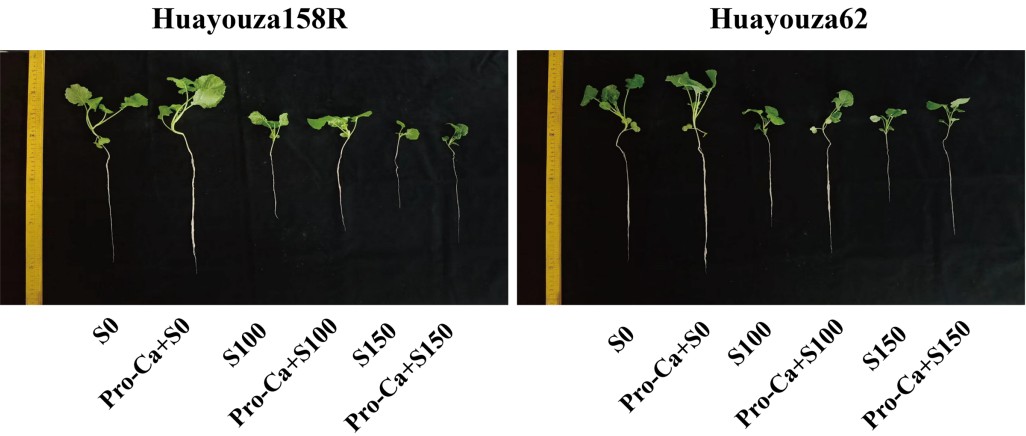

**Figure 1 Visualization of the growth of two rapeseed seedlings under each treatment.** Effect of Pro-Ca priming on the morphology of rapeseed under salt stress.

**Figure 2 Effect of Pro-Ca priming on the growth of oilseed rapeseed under salt stress.** (A) Plant height, (B) stem thickness, (C) leaf area, (D) dry weight, (E) seedling strength index. Uppercase letters indicate significant differences between Huayouza 158R treatments and lowercase letters indicate significant differences between Huayouza 62 treatments. The different letters are significant differences according to Duncan's new multiple range test ($P < 0.05$) based on one-way ANOVA.

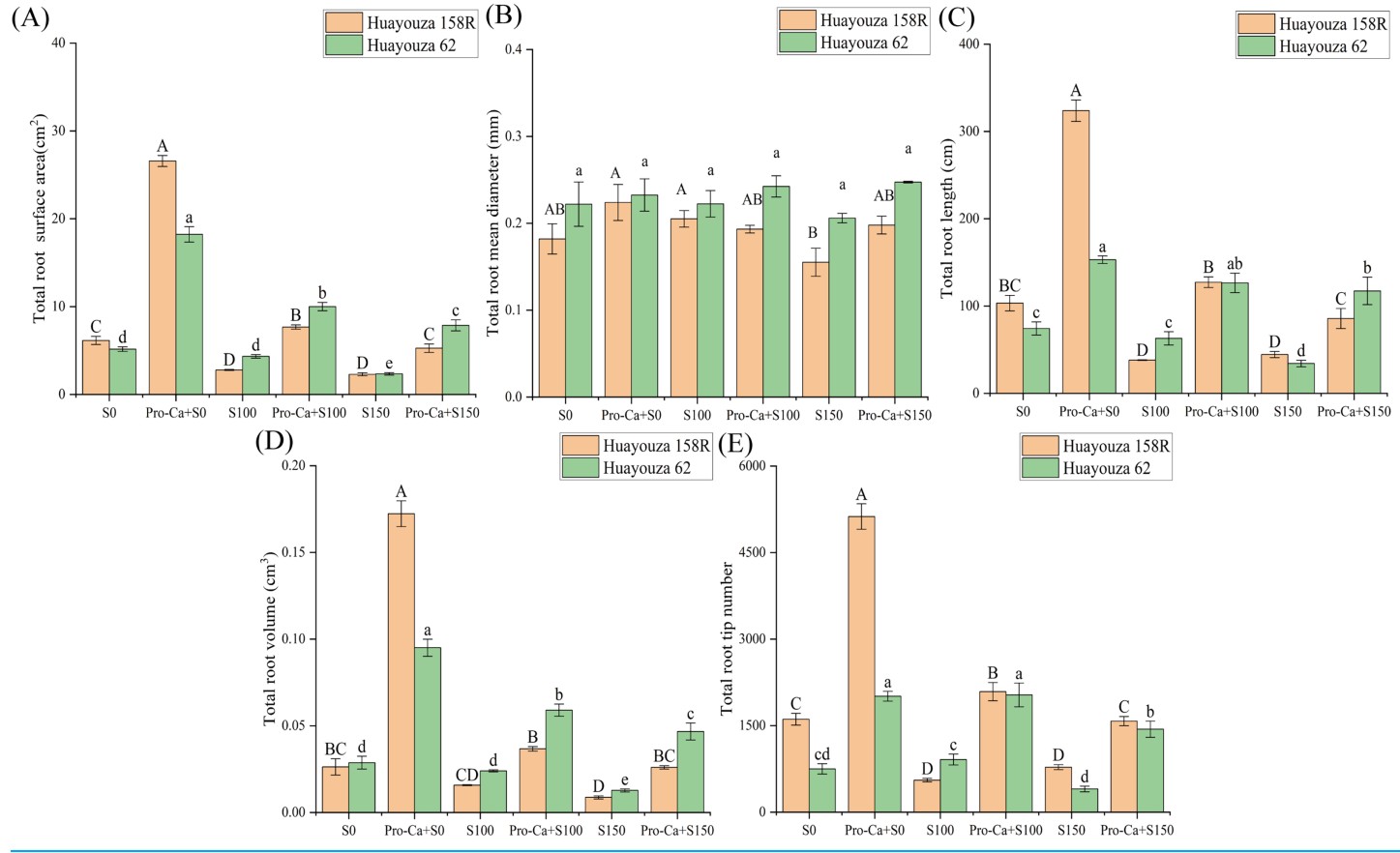

**Figure 3** **Effect of Pro-Ca priming on root growth of rapeseed under salt stress.** (A) Total root surface area, (B) total root mean diameter, (C) total root length, (D) total root volume, (E) total root tip number. Uppercase letters indicate significant differences between Huayouza 158R treatments and lowercase letters indicate significant differences between Huayouza 62 treatments. The different letters are significant differences according to Duncan's new multiple range test ($P < 0.05$) based on one-way ANOVA.

root tip number increased by 131.03%, 100.84%, 145.83%, and 122.81% in Huayouza 62, respectively (Figs. 3A, 3C, 3D and 3F). The study showed that Pro-Ca may alleviate the inhibitory effect of salt stress on the root system.

## Effect of Pro-Ca priming on photosynthetic indexes of rapeseed leaves under salt stress

Salt stress inhibited photosynthesis in rapeseed seedlings, and the inhibition increased with the increase of salt stress. In Huayouza 158R, $Pn$, $Gs$, $Ci$, $Tr$, and $\Phi PSII$ were significantly reduced under S100 and S150 treatments (Figs. 4A–4E). At S100, $Pn$, $Gs$, $Ci$, $Tr$, and $\Phi PSII$ were significantly reduced by 8.00%, 14.53%, 3.80%, and 16.50%, respectively. S150 decreased by 16.63%, 28.61%, 3.00%, and 28.27%, respectively (Figs. 4A–4E). In Huayouza62, $Gs$ was significantly decreased by 16.88% under S100 treatment, and $Pn$, $Ci$, $Tr$, and $\Phi PSII$ were suppressed, but the changes were not significant. Compared with S0, $Pn$, $Gs$, and $Tr$ of S150 decreased significantly by 11.62%, 38.89%, and 17.90%, respectively, and the changes of $Ci$ and $\Phi PSII$ were not significant (Figs. 4A–4E). SPAD content tended

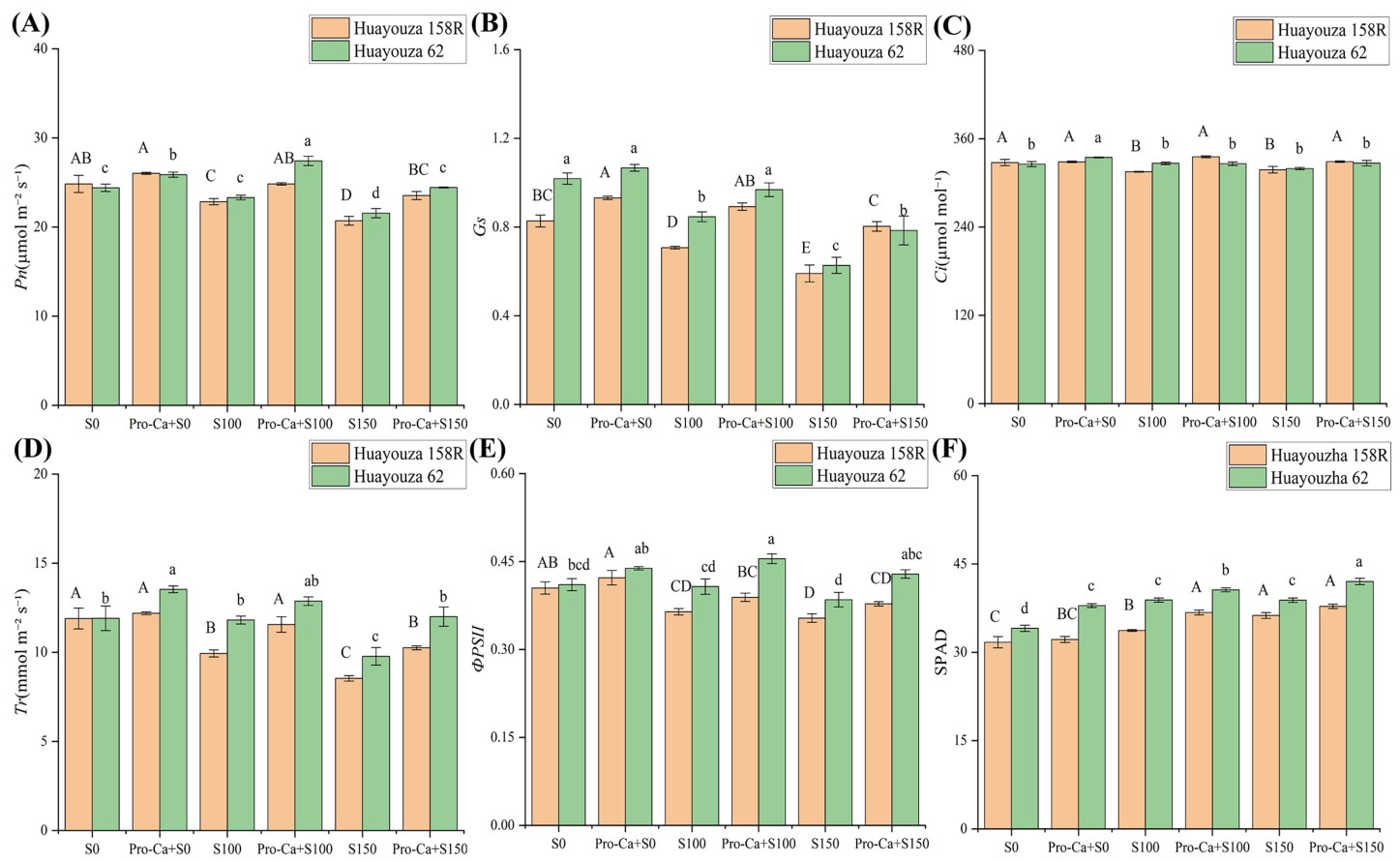

**Figure 4 Effect of Pro-Ca priming on photosynthetic indexes of rapeseed leaves under salt stress.** (A) *Pn*, Net photosynthetic rate, (B) *Gs*, stomatal conductance, (C) *Ci*, intercellular CO2 (D) *Tr*, transpiration rate (E) *ΦPSII*, actual photochemical quantum efficiency (F) SPAD, relative chlorophyll content. Uppercase letters indicate significant differences between Huayouza 158R treatments and lowercase letters indicate significant differences between Huayouza 62 treatments. The different letters are significant differences according to Duncan's new multiple range test (*P* < 0.05) based on one-way ANOVA.                                   

to increase in the salt stress treatment compared to the unstressed. After Pro-Ca primed under salt stress, the SPAD content of Huayouza 62 increased significantly, while that of Huayouza 158R increased significantly only at S100 (Fig. 4F).

Pro-Ca attenuated the inhibition of photosynthesis in rapeseed seedlings by salt stress. Huayouza 158R, *Gs* increased significantly in Pro-Ca+S0 treatment, and promoted *Pn*, *Ci*, *Tr*, and *ΦPSII*, but did not reach a significant level. Pro-Ca+S100 and Pro-Ca+S150 treatments increased *Pn*, *Gs*, *Ci*, and *Tr* significantly, and *ΦPSII* did not reach a significant level. It increased *Pn*, *Gs*, *Ci*, and *Tr* by 8.76%, 26.15%, 2.38%, 16.38%, and 13.70%, 36.03%, 3.44%, and 20.14%, respectively (Figs. 4A–4E). After Huayouza 62, *Pn*, *Ci*, and *Tr* of Pro-Ca+S0 increased significantly by 6.05%, 2.80%, and 13.63%, while *Gs* and *ΦPSII* did not reach significant levels. Pro-Ca+S100 significantly increased *Pn*, *Gs*, and *ΦPSII* by 17.63%, 14.42%, and 11.66%, and contributed to *Ci* and *Tr*, but not significantly. Under the treatment of Pro-Ca+S150, *Pn*, *Gs*, *Tr*, and *ΦPSII* were significantly increased by 13.32%, 25.09%, 22.78%, and 11.34%, respectively, and *Ci* was increased but not significantly

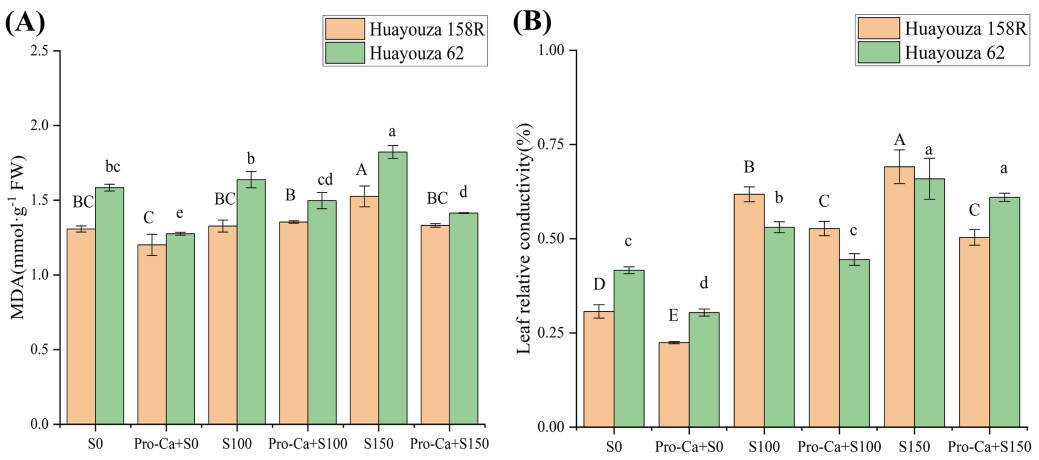

**Figure 5 Effect of Pro-Ca priming on cell damage and membrane stability in rapeseed leaves under salt stress.** (A) MDA, malondialdehyde, (B) leaf relative conductivity. Uppercase letters indicate significant differences between Huayouza 158R treatments and lowercase letters indicate significant differences between Huayouza 62 treatments. The different letters are significant differences according to Duncan's new multiple range test ($P < 0.05$) based on one-way ANOVA.

(Figs. 4A–4E). The study showed that Pro-Ca could increase $Gs$ and $Tr$ in oilseed rape, thus promoting $Pn$ to alleviate the inhibitory effect of salt stress.

## Effect of Pro-Ca priming on cell damage and membrane stability in rapeseed leaves under salt stress

Salt stress disrupted the cell membrane structure of rapeseed, resulting in an increase in MDA and relative osmotic pressure of cellular electrolytes (Fig. 5). The S100 of Huayouza 158R showed a non-significant increase in MDA content and a significant increase in relative conductivity by 101.31%. The S150 showed a significant increase in MDA and relative conductivity by 16.67% and 125.14%, respectively. The S100 of Huayouza 62 showed a non-significant increase in MDA and a significant increase in relative conductivity by 27.38%. S150 showed a significant increase in MDA and relative conductivity by 15.07% and 58.29%, respectively. The increase in cell membrane permeability but not significant increase in MDA content at 100 mM NaCl in the experiment indicated that salt stress at this point was somewhat detrimental to rapeseed but within tolerance. At 150 mM NaCl, the damage to the cells reached a significant level (Fig. 5).

Pro-Ca was able to reduce the cellular damage caused by salt stress. The MDA content of Huayouza 158R Pro-Ca+S100 did not change much, while the relative conductivity was significantly reduced by 14.73%. The MDA content and relative conductivity of Pro-Ca+S150 were significantly reduced by 12.70% and 27.12%, respectively. For Huayouza 62, the MDA content and relative conductivity of Pro-Ca +S100 were significantly decreased by 8.53% and 16.15%, respectively. The MDA content of Pro-Ca+S150 was significantly decreased by 22.41%, and the relative conductivity was not significantly decreased (Fig. 5).

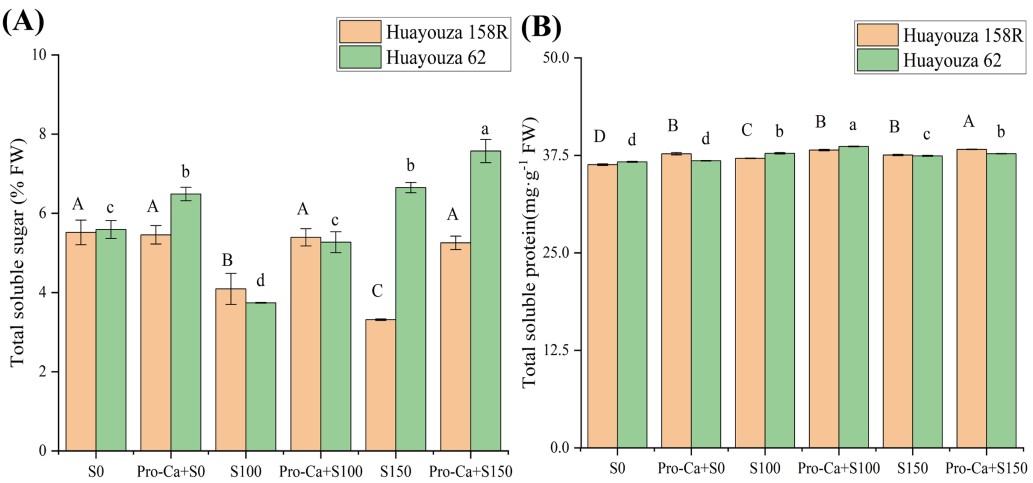

**Figure 6 Effect of Pro-Ca triggering on osmoregulatory nodal substances in rapeseed seedlings under salt stress.** (A) total soluble sugars, (B) total soluble protein. Uppercase letters indicate significant differences between Huayouza 158R treatments and lowercase letters indicate significant differences between Huayouza 62 treatments. The different letters are significant differences according to Duncan's new multiple range test ($P < 0.05$) based on one-way ANOVA.

## Effect of Pro-Ca priming on osmoregulatory nodal substances in rapeseed seedlings under salt stress

Under salt stress, the total soluble sugar content of rapeseed showed a decreasing trend and the total soluble protein content showed an increasing trend. Huayouza 158R showed a significant decrease of 39.95% in the total soluble sugar content and a significant increase of 3.4% in the total soluble protein content under the S150 treatment (Fig. 6). The total soluble sugar content of Huayouza 62 decreased and then increased under salt stress (Fig. 6A). Under S150 treatment, the total soluble sugar content increased by 18.92% and the total soluble protein content increased by 2.07% (Fig. 6).

The Pro-Ca primer may promote the synthesis of osmoregulatory substances. In Huayouza 158R, the total soluble sugar content of Pro-Ca+S0 increased significantly while the total soluble protein content did not. Under salt stress, total soluble sugars and total protein contents were significantly increased in Pro-Ca stimulated treatments (Fig. 6). In Huayouza 62, the total soluble protein content of Pro-Ca+S0 was significantly increased and the total soluble sugar content did not reach a significant level. The total soluble sugar and total protein contents were significantly increased in Pro-Ca induced treatments under salt stress conditions (Fig. 6). This suggests that Pro-Ca induction promotes the synthesis of osmoregulatory substances in response to salt stress.

## Effect of Pro-Ca priming on antioxidant enzyme activity in rapeseed seedlings under salt stress

Antioxidant enzyme activities increased under salt stress. In Huayouza 158R, SOD, POD, and CAT were significantly increased in S100, APX was significantly decreased, and PAL was not significantly changed (Fig. 7). In S150, there was a significant increase in SOD and

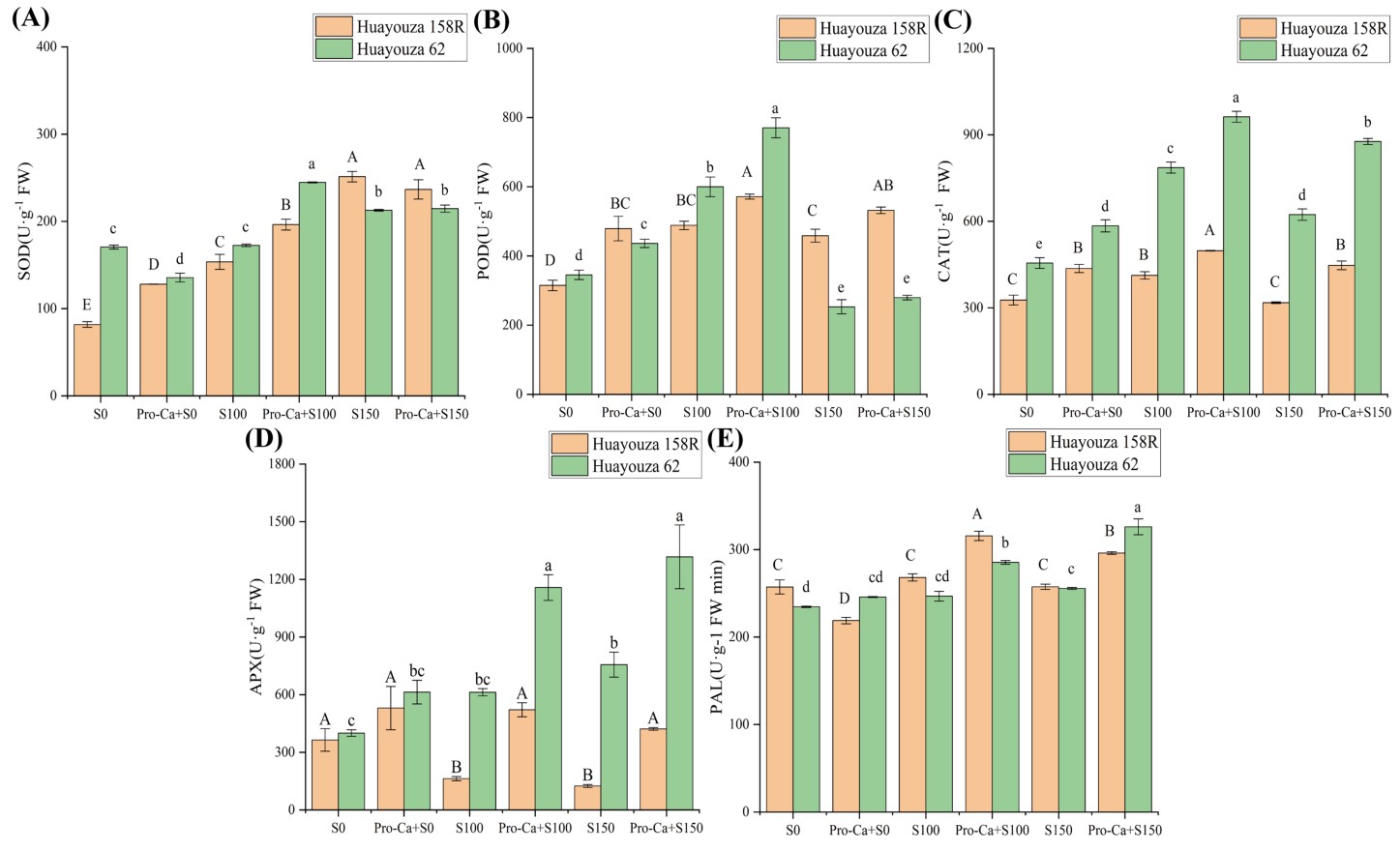

**Figure 7 Effect of Pro-Ca priming on antioxidant enzyme activity in rapeseed seedlings under salt stress.** (A) SOD, superoxide dismutase (B) POD, peroxidase (C) CAT, catalase (D) APX, ascorbate peroxidase (E) PAL, phenylalnine ammonialyase. Uppercase letters indicate significant differences between Huayouza 158R treatments and lowercase letters indicate significant differences between Huayouza 62 treatments. The different letters are significant differences according to Duncan's new multiple range test ($P < 0.05$) based on one-way ANOVA.

POD and 65.74% decrease in APX (Figs. 7A, 7B and 7D). In Huayouza 62, there was a significant increase in POD and CAT in S100 (Figs. 7B and 7C). In S150, there was a significant increase in SOD, CAT, APX, and PAL and a significant decrease of 26.76% in POD (Fig. 7).

Pro-Ca triggering significantly promoted the synthesis of antioxidant enzymes. Huayouza 158R, Pro-Ca+S100 treatment significantly increased antioxidant enzyme activities by 27.78%, 17.02%, 20.86%, 43.17%, and 22.71%, respectively. POD, CAT, APX, and PAL were significantly increased by 15.97%, 40.82%, 237.84%, and 14.99%, respectively, while SOD was significantly decreased by 5.84% in Pro-Ca+S150 as compared to S150 (Fig. 7). In Huayouza 62, Pro-Ca+S0 showed a significant decrease of 20.53% in SOD and a significant increase of 26.35% and 28.40% in POD and CAT, respectively (Figs. 7A–7C). Pro-Ca+S100 treatment significantly increased the antioxidant enzymes by 41.81%, 28.45%, 22.42%, 88.99%, and 15.64%, respectively (Fig. 7). With Pro-Ca+S150, POD, CAT, APX, and PAL were significantly increased by 10.52%, 40.69%, 74.19%, and

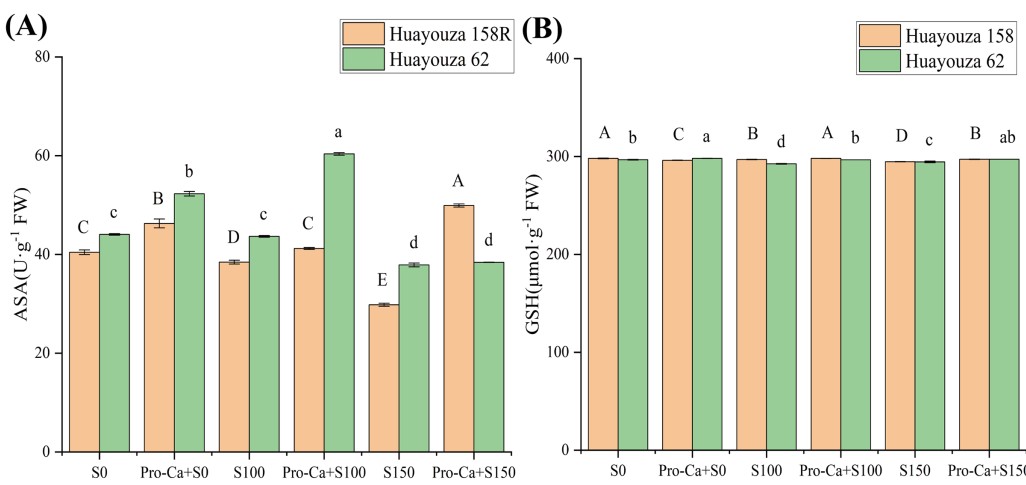

**Figure 8 Effect of Pro-Ca priming on ASA and GSH contents of rapeseed seedlings under salt stress.**
(A) ASA, ascorbic acid (B) GSH, glutathione. Uppercase letters indicate significant differences between Huayouza 158R treatments and lowercase letters indicate significant differences between Huayouza 62 treatments. The different letters are significant differences according to Duncan's new multiple range test ($P < 0.05$) based on one-way ANOVA.

27.57%, respectively (Figs. 7B–7E). The experiment showed that Pro-Ca triggering was effective in promoting the synthesis of antioxidant enzymes.

## Effect of Pro-Ca priming on ASA and GSH contents of rapeseed seedlings under salt stress

Salt stress inhibited the synthesis of ASA and GSH. In Huayouza 158R, the ASA and GSH contents were significantly reduced under the S100 and S150 treatments. In Huayouza 62, the GSH content was significantly reduced in the S100 treatment, while ASA and GSH content was significantly reduced in the S150 treatment (Fig. 8).

Pro-Ca elicitation promotes the synthesis of ASA and GSH. In Huayouza 158R, Pro-Ca +S0 treatment showed a significant increase in ASA and a significant decrease in GSH. Pro-Ca+S100 treatment showed a significant increase in ASA and GSH. Pro-Ca+S150 showed a significant increase in ASA and GSH. Huayouza 62, Pro-Ca+S0 treatment showed a significant increase in ASA and GSH. ASA and GSH were significantly increased in Pro-Ca+S100 treatment. GSH increased significantly in Pro-Ca+S150 treatment, while the change in ASA was only 1.38%, which did not reach a significant level (Fig. 8).

## DISCUSSION

In this study, seedlings grew slowly, and plants were weak due to the adverse effects of salt stress; the development of the root system was significantly reduced. To investigate the effects of NaCl stress on oilseed rape seedlings and ways to improve the salt tolerance of seedlings, exogenous chemical regulation of plants was carried out by special treatments of growth regulators (primed treatments). After Pro-Ca regulation was primed, it could not only alleviate salt damage through antioxidant defense but also maintain the photosynthetic capacity of oilseed rape under salt stress, which significantly alleviated the inhibition of salt stress on the growth and development of oilseed rape (Fig. 9). Several

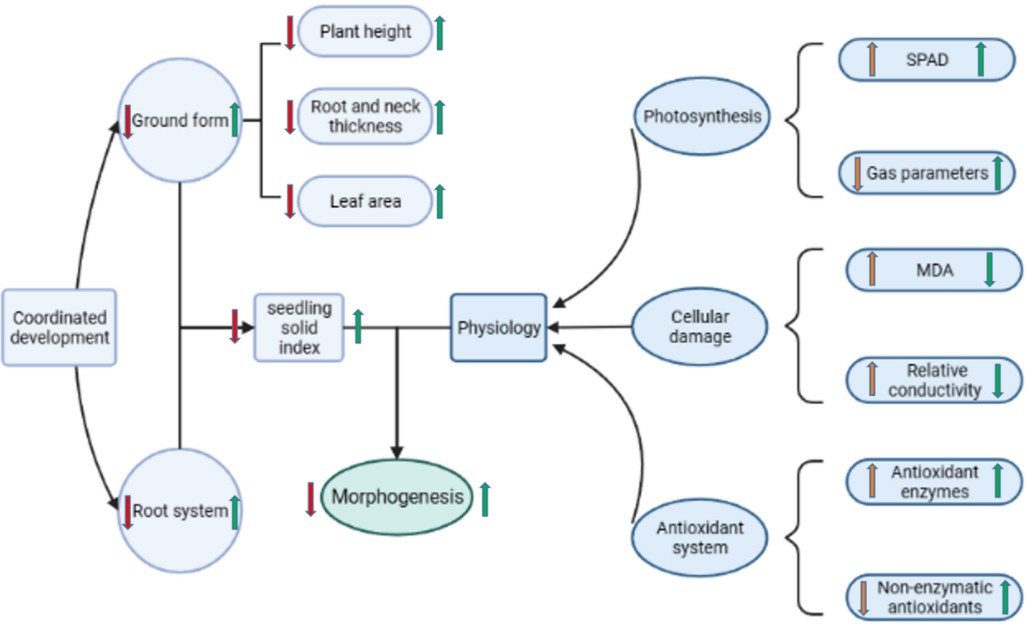

**Figure 9 Diagram of the regulatory mechanism of Pro-Ca priming on rapeseed under salt stress.** Red arrows indicate the changes in each metric in salt stress relative to no stress; green arrows indicate the changes in each metric in the Pro-Ca primed treatment relative to the unprimed treatment under salt stress.

scholars have shown that foliar spraying of Pro-Ca can regulate crop growth and alleviate salt stress injury (*Feng et al., 2021*; *Zhang et al., 2023*). Unlike these scholars, our study showed that Pro-Ca primed could also effectively promote the root growth of oilseed rape. Secondly, this study used a more efficient treatment with less dosage of Pro-Ca and achieved significant results.

## Pro-Ca priming promotes morphogenesis of rapeseed seedlings under salt stress

The results of this study showed that salt stress inhibits the growth of rapeseed, leading to thinning of the stem thickness, significant reduction of leaf area, and blockage of organic matter synthesis (Figs. 1 and 2). The reason may be that salt stress destroys cellular structures, leading to lipid peroxidation and degradation of cellular membrane systems, which can have serious negative effects on its growth (*Li et al., 2022*; *Popova, Borisova & Vasilev, 2023*). This is consistent with our findings that salt stress leads to weakened cellular functions. The growth of Huayouza 158R was inhibited to varying degrees at different salt concentrations, and all growth indices decreased significantly at high salt concentrations. However, the inhibition rate of Huayouza 62 did not change significantly, indicating that Huayouza 62 has a stronger tolerance and is more suitable for growth in high salinity areas (Fig. 5). The results are in agreement with those of *Gogna & Bhatla (2020)*, who described that different varieties have different sensitivities under salt stress. The seedling strength index reflects the robustness of plant growth (*Liu et al., 2015*); the seedling strength index of Huayouza 158R decreased significantly as the degree of salt

stress increased. However, there was no significant change in Huayouza 62 (Fig. 2E), suggesting that Huayouza 62 grows more robustly and is more resistant to salt stress.

Salt stress significantly inhibited the growth and development of the root system (*Li et al., 2023*). The change in mean diameter of the root was noteworthy and significantly decreased at treatment S150 for Huayouza 158R. The non-significant changes of Huayouza 62 under salt stress indicates that the difference in salt tolerance between the two varieties may be due to the difference in average root diameter (Fig. 3B). The robust root system of Huayouza 62 may contribute to the improvement of soil water and nutrient utilization. Pro-Ca priming may reflect the positive response of the root system to environmental changes.

Our findings showed that Pro-Ca priming significantly alleviated salt stress injury of the root system and improved the nutrient uptake capacity of rapeseed by promoting the root surface area and the number of root tips (Figs. 3A and 3E). Our finding corroborate those of *Wu et al. (2019)*, whose field study indicated that under salt stress, Pro-Ca primers promoted vigorous seedling growth attributed to several aspects such as alleviating salt stress injury, and promoting leaf photosynthetic capacity and root system nutrient uptake. Pro-Ca priming resulted in an increased root volume and improved the overall growth and development of rapeseed seedlings (Fig. 3D); this effect may be due to the enhanced nutrient uptake and transportation capacity of rapeseed (*Fritz & Ehwald, 2013*). The increased root system provided more nutrients to the aboveground plant parts under salt stress, alleviating the water stress and ion toxicity caused by salt stress, resulting in more robust plants, increased tolerance to salt stress, and elevated seedling strength index.

## Pro-Ca priming promotes the photosynthetic capacity of rapeseed seedlings under salt stress

Our study showed that salt stress reduced the photosynthetic parameters of both varieties (Fig. 4), which is consistent with the performance of most plants under salt stress (*Sehar, Masood & Khan, 2019*). However, Huayouza 62 maintained a higher photosynthetic rate under salt stress (Fig. 4A). *Ning et al. (2018)*'s analysis indicated that salt-tolerant types had a higher photosynthetic rate than salt-sensitive ones under salt stress. The growth of Huayouza 62 was inhibited under S100 conditions, when only *Gs* decreased significantly, suggesting that the opening and closing of stomata is the main cause of photosynthesis under salt stress (Fig. 4B). Similar results were observed by *Mohamed et al. (2020)*, who demonstrated that leaf stomatal conductance was highly correlated with salt tolerance. Pro-Ca primers under salt stress maintained higher *Tr* and *Gs*, which favored leaf stomatal opening and supported gas exchange, and improved the electron transport capacity and $CO_2$ fixation efficiency of the chloroplast light energy complex, thus promoting photosynthesis (Figs. 4B and 4D). The accumulation of net photosynthetic products in the plant ensures the supply of nutrients, which might be the reasons why Pro-Ca primers promote salt tolerance in rapeseed.

Prolonged salt stress leads to a reduction in the biosynthesis of chlorophyll protein-lipid complexes, which affects the photosynthetic system and leads to retarded plant growth (*Guo et al., 2019*; *Xue et al., 2021*; *Yin et al., 2019*). However, no uniform conclusion has

emerged regarding the changes in photosynthetic pigments under salt stress (*Kumar et al., 2021*), and other studies have pointed out that the increase in chlorophyll content is transient, increasing and then decreasing with the nonstop action of salt stress (*Gadelha et al., 2021*). We showed that SPAD content tended to increase with the intensification of salt stress (Fig. 4F). Salt stress may inhibit leaf growth, resulting in intracellular chloroplast accumulation; this effect may be related to the variety of rapeseed. Carotenoids may be involved in oxidative stress and the dark green color of leaves under salt stress may be related to the involvement of carotenoids in oxidative stress decomposition (*Ali et al., 2017*). Chlorophyll synthesis under salt stress reflects a plant's salt tolerance (*Soliman et al., 2020*). Significant differences in SPAD content were noted between the two species after Pro-Ca priming, and there was a substantial increase in SPAD content in Huayouza 62 at high salt stress, suggesting that Pro-Ca priming may promote chlorophyll synthesis under salt stress (Fig. 4F). The present study suggested that cellular respiration may be enhanced under salt stress stimulation, resulting in an increase in SPAD content but a decrease in *Pn*. Our results were similar to those of *Gong et al. (2020)*, who found that cellular respiration was significantly enhanced under salt stress. The application of Pro-Ca alleviated the unfavorable effects of salt stress and increased the relative SPAD content and various photosynthetic traits of rapeseed. Pro-Ca priming induced the plants to synthesize chlorophyll and promoted stomatal conductance, the transpiration rate, and other indexes, which led to a significant increase in the net photosynthetic rate.

## Pro-Ca priming promotes the synthesis of osmoregulatory substances in rapeseed seedlings under salt stress

The significant accumulation of harmful ions under salt stress produces osmotic stress. It inhibits the nutrient uptake of plants and causes the accumulation of soluble osmoregulatory substances in the cells to reduce the intracellular osmotic potential and increase the water uptake of the cells (*Shabala et al., 2020*). Our study showed that the content of soluble sugars increased in response to this osmotic stress. The soluble protein content differed between the two rapeseed varieties, mainly synthesizing soluble sugars to reduce the cellular osmotic potential under low salt stress (Fig. 6). Huayouza 62 improved the synthesis of total soluble protein content to enhance tolerance against salt stress under high salt stress (Fig. 6B). This is consistent with the results of *Abdel Latef et al. (2020)*. The higher soluble protein content regulates the osmotic pressure and may also represent the activation of additional physiological and biochemical responses under high salt stress. The soluble sugars and soluble proteins exhibit a compensatory effect. When soluble sugars increase or decrease in relation, soluble proteins show the opposite trend. The synthesis of osmoregulatory substances increased significantly after Pro-Ca priming, suggesting that Pro-Ca priming increased the physiological activity of rapeseed. Pro-Ca can regulate the content of soluble sugars and soluble proteins inside and outside the cell, maintain the relative balance of osmotic potential, and facilitate resistance to salt stress.

## Pro-Ca priming the regulation of the antioxidant system in rapeseed seedlings under salt stress

Salt stress induces the production of large amounts of reactive oxygen species (ROS), which disrupts the relative balance of intracellular ROS. High concentrations of ROS cause cellular damage (*Cheng et al., 2021*) and MDA content and cell membrane permeability may reflect the extent of cell membrane damage under salt stress and are used to assess cell injury (*Nie et al., 2023*). Our study illustrated that the high MDA content of Huayouza 62 may have a high membrane lipid peroxidation base under normal conditions, however, this effect needs to be clarified (Fig. 5A). Our results are supported by *Poursakhi, Razmjoo & Karimmojeni (2019)*, who found that Pro-Ca priming significantly reduced the MDA content under salt stress, probably because Pro-Ca priming reduces the electron-skimming of cell membrane lipids by oxygen radicals, attenuates the oxidative degradation of membrane lipids under adversity, and reduces cell damage. The relative conductivity of the leaves decreased significantly after Pro-Ca elicitation (Fig. 5B), which indicated that Pro-Ca elicitation protected the cell membrane system and maintained its integrity, which was closely related to the improvement of antioxidant capacity by Pro-Ca priming.

Plants rely on antioxidant systems to eliminate excess reactive oxygen species to maintain relative homeostasis and antioxidant systems are divided into antioxidant enzymes and antioxidants (*Khan et al., 2020*). This study showed that rapeseed under salt stress promoted the production of enzyme syntheses such as SOD, POD, and CAT to cope with the excess ROS produced under salt stress (Figs. 7A–7C). It was found that under high salt stress, Pro-Ca elicitation did not significantly promote SOD and POD enzyme activities (Figs. 7A and 7B). Huayouza 62 enzyme activity was higher than Huayouza 158R, showing higher resistance to salt stress. The actions of SOD and POD peaked at low salt stress after Pro-Ca priming. The effects of other antioxidant enzymes increased with increasing salt stress, suggesting that the mechanism of Pro-Ca primed to promote the antioxidant system of rapeseed varied under different salt stresses. Under low salt stress, all antioxidant enzymes were significantly promoted, and under high salt stress, the synthesis of CAT, APX, and PAL enzymes was mainly promoted to cope with the oxidative stress of salt stress (Fig. 7). Interestingly, APX activity increased significantly after Pro-Ca elicitation, and APX activity increased with the increase of salt stress after elicitation, suggesting that Pro-Ca priming has a non-negligible promotion effect on APX synthesis (Fig. 7D). The secondary phenolic metabolic synthesis pathway is activated under stress conditions, and PAL is a critical enzyme that encourages disease and stress resistance in plants (*Gholizadeh & Kohnehrouz, 2010*). The PAL content tended to increase but did not reach significant levels under salt stress. However, after Pro-Ca priming, PAL increased considerably, suggesting that Pro-Ca priming significantly promotes PAL ase activity (Fig. 7E). This effect may enhance secondary phenol metabolism in plants and thus improve the antioxidant capacity of rapeseed. ASA and GSH are prevalent plant antioxidants in various redox reactions. Our results were consistent with *Hasanuzzaman et al. (2019)*. Salt stress inhibits the ASA-GSH cycle, significantly decreasing ASA and GSH content (Fig. 8). Current findings demonstrated that Pro-Ca priming promotes the

synthesis of ASA and GSH for ROS elimination and the synthesis of large amounts of substrates also promotes the ASA-GSH cycle. Our results are in agreement with those of *Jiang et al. (2020)*, who reported that ASA-GSH alleviates oxidative stress in rapeseed under salt stress. Thus, the ability of Pro-Ca to eliminate reactive oxygen species is remarkable, and improves the ability of the named pathways to maintain the relative balance of reactive oxygen species.

## CONCLUSION

In conclusion, our experimental results demonstrated that the Pro-Ca primer could significantly improve the antioxidant systems, including SOD, POD, and CAT under salt stress. Other enzyme activities and the ASA-GSH cycle were also improved and Pro-Ca application effectively reduced the membrane lipid peroxidation damage. It also promoted leaf photosynthesis-related parameters, increased the relative chlorophyll content, and promoted the synthesis of osmoregulatory substances to cope with salt stress. The results also showed that the two rapeseed varieties differed greatly under salt stress, with Huayouza 62 showing higher salt tolerance due to its stronger ROS elimination system and photosynthetic capacity. Therefore, the Pro-Ca primer effectively promoted salt tolerance in rapeseed and improved the sustainability of rapeseed cultivation in saline soils. This study suggested that Pro-Ca primer had a good potential in promoting salt tolerance in plants.

### Funding

Funding was received from the Funding Sources National 13th Five-Year Plan Key R&D Program Project (2019YFD1002205), the Guangdong Provincial Department of Education General College Innovation Team Project (2021KCXTD011), and the Guangdong Provincial Department of Education Graduate Student Innovation Forum (2022XSLT036). The funders had no role in study design, data collection and analysis, decision to publish, or preparation of the manuscript.

### Grant Disclosures

The following grant information was disclosed by the authors:
Funding Sources National 13th Five-Year Plan Key R&D Program Project: 2019YFD1002205.
Guangdong Provincial Department of Education General College Innovation Team Project: 2021KCXTD011.
Guangdong Provincial Department of Education Graduate Student Innovation Forum: 2022XSLT036.

### Competing Interests

Naijie Feng and Dengfeng Zheng are employed by the National Innovation Center for Salt-tolerant Rice Technology. The authors declare that the research was conducted in the

absence of any commercial or financial relationships that could be construed as a potential conflict of interest.

## Author Contributions

- Peng Deng conceived and designed the experiments, performed the experiments, analyzed the data, prepared figures and/or tables, authored or reviewed drafts of the article, experimental design, and approved the final draft.
- Aaqil Khan performed the experiments, prepared figures and/or tables, authored or reviewed drafts of the article, and approved the final draft.
- Hang Zhou analyzed the data, prepared figures and/or tables, authored or reviewed drafts of the article, and approved the final draft.
- Xutong Lu performed the experiments, prepared figures and/or tables, and approved the final draft.
- Huiming Zhao performed the experiments, prepared figures and/or tables, and approved the final draft.
- Youwei Du analyzed the data, prepared figures and/or tables, and approved the final draft.
- Yaxin Wang analyzed the data, prepared figures and/or tables, and approved the final draft.
- Naijie Feng conceived and designed the experiments, prepared figures and/or tables, authored or reviewed drafts of the article, test proofreading, and approved the final draft.
- Dianfeng Zheng conceived and designed the experiments, prepared figures and/or tables, authored or reviewed drafts of the article, and approved the final draft.

## Data Availability

The raw measurements and analysis are available in the Supplemental Files.

## Supplemental Information

Supplemental information for this article can be found online at http://dx.doi.org/10.7717/peerj.17312#supplemental-information.

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
