# Peer review of "Application of prohexadione-calcium priming affects *Brassica napus* L. seedlings by regulating morph-physiological characteristics under salt stress"

_PeerJ, doi:10.7717/peerj.17312_

## Round 0.1 · original submission · Minor Revisions

Dear Authors,

Please improve the quality of your article as advised by the Reviewers. Pay attention to grammatical errors, reference citations, and discussion.

Regards

**Language Note:** The Academic Editor has identified that the English language must be improved. PeerJ can provide language editing services - please contact us at [email protected] for pricing (be sure to provide your manuscript number and title). Alternatively, you should make your own arrangements to improve the language quality and provide details in your response letter. – PeerJ Staff

Reviewer 1 ·

Basic reporting

Although MS “Application of prohexadione-calcium priming eûect (Brassica napus L.) seedlings by regulating morph-physiological characteristics under salt stress”-is well written but the expression of English is not good and there is room for revisions. e.g., “Analysis of the data showed that salt stress induced morphophysiological traits” can be expressed as “Analysis of the data showed that salt stress induced alterations in morphophysiological traits”.
Moreover, there is lack of proper punctuations at many places like;
Seeds of oilseed rape varieties Huayouza 158R and Huayouza 62 were used as plant material in this study
Should be written as
Seeds of oilseed rape varieties, Huayouza 158 R and Huayouza 62, were used as plant material in this study
Conclusion need to be rewritten. There should be no introduction like this
Salt stress disrupts the cell integrity of rapeseed seedlings, inhibits photosynthesis and antioxidant systems, and negatively affects the morphogenesis of rapeseed seedlings.
All scientific words need to be italics

Experimental design

ok

Validity of the findings

Figure 3 looks that authors took the pics of same plants but claimed 2 different varieties

Additional comments

NA

Reviewer 2 ·

Basic reporting

there are typos and English needs improvement. Lines 46,47, 0.34x109 should be in standard form, line 93, no seeds are in each petri dish. Similarly, there are more typing mistakes.
References are sufficient, a discussion needs more justification.

Experimental design

The experiment design is well-defined.

Validity of the findings

Findings are elaborated well.

Additional comments

Minor revision

·

Basic reporting

.

Experimental design

.

Validity of the findings

.

Additional comments

The article entitled "Application of prohexadione-calcium priming eûect (Brassica napus L.) seedlings by regulating morph-physiological characteristics under salt stress" is well-compiled manuscript, and the authors compared the differences of morphophysiological characteristics, and osmoregulatory and antioxidant activity between different oilseed rape varieties. In general, the results are innovative, significant and useful for the research on investigating the mechanims of Pro-Ca sed priming on rapseed development in response to salt stress. However, several technical issues should be addressed first.
(1)The writing of the manuscript needs improvement, and some grammatical and spelling errors still exist. Suggest to invite someone native to help with the polish.
(2)We noticed that this manuscript was similar as another article entitled as “Effects of prohexadione calcium spraying during the booting stage on panicle traits, yield, and related physiological characteristics of rice under salt stress”, while could the authors illustate the innovation and novelty ?
(3)The references were not uniform, as some ones lack the issue number, and some ones use the full names of the cited journal, while some not.The references were not uniform, as some ones utilized all the capital in the title.
(4)The quality and resolution of figures should enhance, and suggest to add the table to prove the figure 3 result.

---

## Round 0.2 · Major Revisions

Dear Authors,

Please improve your article as advised by the Reviewers

Regards

**Language Note:** The review process has identified that the English language must be improved. PeerJ can provide language editing services - please contact us at [email protected] for pricing (be sure to provide your manuscript number and title). Alternatively, you should make your own arrangements to improve the language quality and provide details in your response letter. – PeerJ Staff

Reviewer 1 ·

Basic reporting

I have again gone through the MS “Application of prohexadione-calcium priming effect (Brassica napus L.) seedlings by regulating morph-physiological characteristics under salt stress”

The English language issues/expression are still there as an example;
Moreover, to check the correlations between morphological traits and physiological traits such as photosynthesis, osmoregulation, and antioxidant defense system of oilseed rape.

The pot experiment used a completely randomized block design using two rapeseed varieties

After priming, the seeds were washed with distilled water, dried, and set aside.

There is no need to mention such details as;
The dry weight was determined using a one-ten-thousandth analytical balance.

Aboveground or above ground?
CO2 or CO2 see this issue at many places
Use equation editor for equations
ASA and GSH and other abbreviations need to be described at 1st mention
Coarseness rhizome, or coarseness of rhizome,
Shoot length or shoot length
There is no sentence connectivity like this at many places;
Favorably maintains the relative balance of osmotic potential, thus maintaining tolerance against salt stress.
Brassica rapa L. L. should not be italics (Although pointed out earlier)

Experimental design

ok

Validity of the findings

Conclusions not corrected as per previous comments

Additional comments

English language expression is too poor although point out earlier during first round

Reviewer 2 ·

Basic reporting

All suggestions are incorporated

Experimental design

Well defined question

Validity of the findings

Conclusion is well statted

·

Basic reporting

The article should include sufficient introduction and background to demonstrate how the work fits into the broader field of knowledge. Relevant prior literature should be appropriately referenced.

Experimental design

Original primary research within Aims and Scope of the journal.

Validity of the findings

All underlying data have been provided; they are robust, statistically sound, & controlled.

Additional comments

Application of prohexadione-calcium priming effect (Brassica napus L.) seedlings by regulating morphphysiological characteristics under salt stress.
In this paper, the effects of prohexadione-calcium on salt stress in Brassica napus were evaluated from growth indexes and physiological and biochemical levels by means of seed priming application. It was found that seed priming had more significant effects on salt stress relief. However, there are a lot of formatting problems, syntax problems, and some incomprehensible places in this paper that need to be further modified. The main problems are as follows:
• Brassica napus L should be bracket free in title.
• The authors discuss seed priming in the introduction and adopt this method in materials and methods. Keeping seed priming in consideration, please replace uninitiated treatment at Line 16 with non-primed. Makes corrections throughout the manuscript to make it easier for the reader.
• Please clearly mention what it means by “significantly improved seedlings” at lines 18–19.
• Please italicize the abbreviation of gas exchange parameters at Line 22: net photosynthetic rate (Pn), stomatal conductance (Gs), transpiration rate (Tr). Carefully check it throughout the manuscript.
• Line63; replaced heading Test materials with appropriate headings.
• What it meant by nutrient soil was "a 3:1 mixture of nutrient soil and sand.”.
• The authors don't mention when treatments with salt (0, 100, and 150 mM) were applied. It is either pre-sowing or during the experiment. How many times did salt stress induce.
• How many seeds per pot were sown?
• Please again replace uninitiated with non-primed.
• Either thinning was performed or not?
• At what age was plant sampling taken?
• Check for space issues. Line 92 was baked in an oven for 30 minutes at 105 °C and dried at 80 °C to a constant weight.
• Check for subscript, intercellular CO2 concentration.
• The author used WinRHIZO root analysis software (Regent Instruments, Quebec, Canada). Is this software being licensed for free?
• Why the author used dot after mol: Line 108, airflow rate of 500 μmol·s−1; Line 121, 10 mL of 0.05 mol·L−1; Line 141, homogenate in 10 mL of 0.05 mol·L−1.
• Line 152 standard error (SE) was plotted in Origin 2021; should be standard error (SE) was plotted in Origin 2021.
• Please write, in lowercase, Line 163: Shoot length.
• Please remove the bracket before Gogna and Bhatla at Line 275. The results are in agreement with those of Gogna and Bhatla (2020).
• Why do Figure 6 (total soluble protein) and Figure 8 (GSH) display non-significant results in all treatments? Justify please.

---

## Round 0.3 · Major Revisions

Dear Authors,

Please revise the title of the manuscript. Moreover, recheck statistical analysis and incorporate the required corrections.

Regards

Reviewer 1 ·

Basic reporting

Title “Application of prohexadione-calcium priming effect Brassica napus L seedlings by regulating morph- physiological characteristics under salt stress”
Should be like this
Application of prohexadione-calcium priming affects Brassica napus L. seedlings by regulating morph- physiological characteristics under salt stress
All scientific words need to be italics including those in references
All references need to be formatted to match journal style

Experimental design

LSD results need to be revisited (rechecked) and shared as supplementary file
Statistical analysis looks wrong/confused. How the analysis was performed? Variety separately or 2 varieties together? Need to mention. Consult some expert.

Validity of the findings

Statistical analysis confused

---

## Round 0.4 · accepted · Accept

Authors have addressed all of the reviewers' comments. I am happy with the current version of the manuscript and recommend it for publication.

Reviewer 1 ·

Basic reporting

The authors have greatly improved
In title morph- physiological should be morph-physiological

Experimental design

The authors have greatly improved

Validity of the findings

The authors have greatly improved